# Predicting metabolic modules in incomplete bacterial genomes with MetaPathPredict

**David Geller-McGrath[1]\*, Kishori M Konwar[2], Virginia P Edgcomb[3], Maria Pachiadaki[1], Jack W Roddy[4], Travis J Wheeler[4]\*, Jason E McDermott[5,6]\***

[1]Biology Department, Woods Hole Oceanographic Institution, Woods Hole, United States; [2]Luit Consulting, Revere, United States; [3]Marine Geology and Geophysics Department, Woods Hole Oceanographic Institution, Woods Hole, United States; [4]R. Ken Coit College of Pharmacy, University of Arizona, Tucson, United States; [5]Computational Sciences Division, Pacific Northwest National Laboratory, Richland, United States; [6]Department of Molecular Microbiology and Immunology, Oregon Health & Science University, Portland, United States

**Abstract** The reconstruction of complete microbial metabolic pathways using 'omics data from environmental samples remains challenging. Computational pipelines for pathway reconstruction that utilize machine learning methods to predict the presence or absence of KEGG modules in incomplete genomes are lacking. Here, we present MetaPathPredict, a software tool that incorporates machine learning models to predict the presence of complete KEGG modules within bacterial genomic datasets. Using gene annotation data and information from the KEGG module database, MetaPathPredict employs deep learning models to predict the presence of KEGG modules in a genome. MetaPathPredict can be used as a command line tool or as a Python module, and both options are designed to be run locally or on a compute cluster. Benchmarks show that MetaPathPredict makes robust predictions of KEGG module presence within highly incomplete genomes.

**\*For correspondence:**
dgellermcgrath@whoi.edu (DG-McG);
twheeler@arizona.edu (TJW);
jason.mcdermott@pnnl.gov (JEMcD)

## Editor's evaluation

This landmark study presents MetaPathPredict, a method that uses deep neural networks to predict the presence or absence of KEGG modules based on annotated features in the genome. The evidence supporting the conclusions is compelling, with a tool that allows for the prediction of KEGG modules in sparse gene sequence datasets.

## Introduction

Microorganisms play a key role in all major biogeochemical cycles on Earth. Accurate and more complete identification of microbial metabolic pathways within genomic data is crucial to understanding their potential activities. This identification of pathways within genomic data, and assessment of their expression, provides important insight into their influence on the chemistry of their environment and their mediation of interactions with other organisms.

In recent decades, the scientific community has significantly advanced its capability to gather and sequence genomes from microorganisms. Key steps in the process of working with isolated genomes, single-amplified genomes (SAGs), or metagenome assembled genomes (MAGs), are identifying genes coding for enzymes that catalyze metabolic reactions and inferring the metabolic potential of the associated organism from these data. These analyses involve comparing protein-coding sequences

with homologous sequences from reference metabolic pathway databases including KEGG (*Kanehisa and Goto, 2000*) and MetaCyc (*Caspi et al., 2018*). Environmental genomes that are recovered from high-throughput sequencing samples vary in their degree of completeness due to numerous factors including limited coverage of low-abundance microbes, composition-based coverage biases, insertion-deletion errors, and substitution errors (*Browne et al., 2020*). Enzymes encoded in genomes are also missed due to limitations in protein annotation methods, that is, undiscovered protein families may be undetected by traditional homology-based methods. This can limit the ability to determine the extent to which these organisms (or communities) can catalyze metabolic reactions and form pathways.

Sequencing biases, novel protein families, and incomplete gene and protein annotation databases lead to missing, ambiguous, or inaccurate gene annotations that create incomplete metabolic networks in recovered environmental genomes. This leads to a challenge in genome analysis: given a set of annotated genes that incompletely covers some known metabolic network, predict whether the network is, in fact, present in that organism (i.e. to predict whether one or more unobserved network components is likely present but unobserved for some reason). Existing algorithms for this metabolic network 'gap filling' largely fall into two categories of approaches: those based on network topology, such as the method utilized by Gapseq (*Zimmermann et al., 2021*), and those that utilize pre-defined KEGG module cutoffs, such as those used by METABOLIC (*Zhou et al., 2022*). Network topology and pathway gene presence/absence cutoffs, however, can lead to underestimation of pathways that are present, particularly in highly incomplete genomes. Parsimony-based algorithms such as MinPath detect gaps in a metabolic network and identify the minimum number of modifications to the network that can be made to activate those reactions (*Ye and Doak, 2009*) its conservative approach, however, can lead to underestimation of the metabolic pathways present in a sample. KEMET (*Palù et al., 2022*) can detect gaps in metabolic pathways by searching unannotated genes in a genome with custom Hidden Markov Models (HMMs) created based on the genome's taxonomy. This approach, however, is limited by the genome taxonomies available in the KEGG GENES database. Other modern tools, such as DRAM (*Shaffer et al., 2020*), provide annotations for metagenomic sequences but do not closely tie these to metabolic pathways. Flux-balance analysis (e.g. Escher-FBA; *Rowe et al., 2018*) utilizes genome-scale metabolic models of organisms and requires experimental growth data for model parameterization; it is not easily applied to incomplete genome data, and the additional required experimental measurements may prohibit application in many use cases.

An emerging set of methods utilize machine learning models to a related problem of classifying microbial organisms' niches based on their genomic features. One such example is a tool called Traitar, which utilizes Support Vector Machines (SVMs) to predict lifestyle and pathogenic traits in prokaryotes based on gene family abundance profiles (*Weimann et al., 2016*). Other recent approaches have used machine learning approaches to train models using eukaryote MAG and transcriptome data to classify trophic mode (autotroph, mixotroph, or heterotroph) based on gene family abundance profiles (*Lambert et al., 2022*; *Alexander et al., 2021*). To our knowledge, there are no existing tools that predict the presence/absence of KEGG metabolic modules via machine learning models trained on gene features of high-quality genomes.

Here, we present 'MetaPathPredict', an open-source tool for metabolic pathway prediction based on a deep learning classification framework. MetaPathPredict addresses critical deficiencies in existing metabolic pathway reconstruction tools that limit the utility and predictive power of 'omics data: it connects manually curated, current knowledge of metabolic pathways from the KEGG database with machine learning methods to reconstruct and predict the presence or absence of KEGG metabolic modules within genomic datasets including bacterial isolate genomes, MAGs, and SAGs.

The models underlying MetaPathPredict contain metabolic reaction and pathway information from taxonomically diverse bacterial isolate genomes and MAGs found in the NCBI RefSeq (*O'Leary et al., 2016*) and Genome Taxonomy (GTDB, *Parks et al., 2022*) databases. The set of metabolic modules from the KEGG database is the basis of the tool's metabolic module reconstruction and prediction. The KEGG database contains metabolic pathway information for thousands of prokaryotic species and strains. KEGG modules are functional units of metabolic pathways composed of sets of ordered reaction steps. Examples include carbon fixation pathways, nitrification, biosynthesis of vitamins, and transporters or two-component systems (see *Supplementary file 1a* for a description of the distribution of modules covered by MetaPathPredict). MetaPathPredict is designed to run on

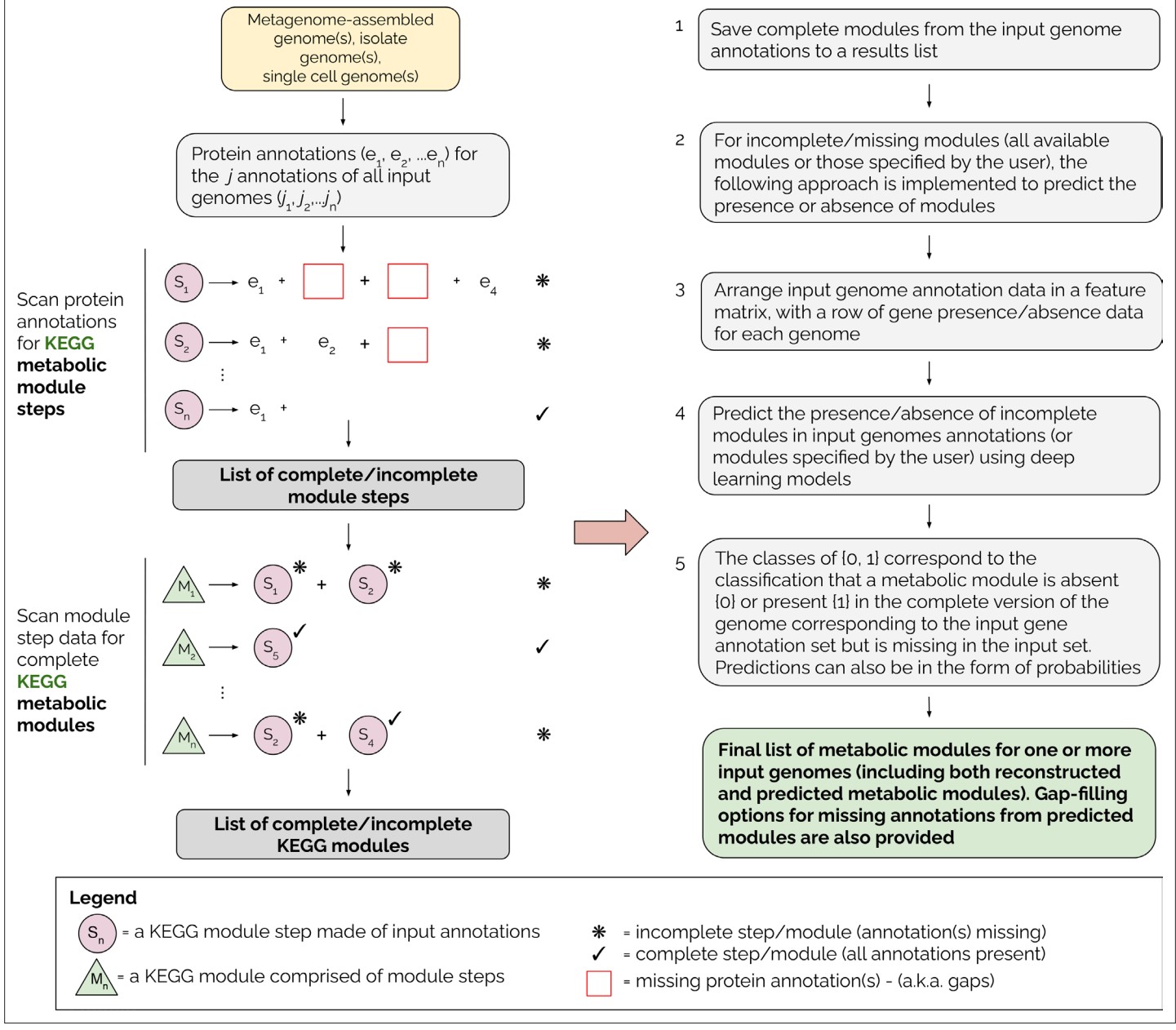

**Figure 1.** Overview of the MetaPathPredict pipeline. Input genome annotations are read into MetaPathPredict as a data object. The data are scanned for present KEGG modules and are formatted into a feature matrix. The feature matrix is then used to make predictions for all incomplete modules (or modules specified by the user). A summary and detailed reconstruction and prediction objects, along with gap-filling options are returned in a list as the final output.

the command-line locally or on a computing cluster and is available as a Python module on GitHub (https://github.com/d-mcgrath/MetaPathPredict).

A detailed overview of the MetaPathPredict pipeline is provided in *Figure 1*. The tool accepts as input gene annotations of one or more (possibly-incomplete) genomes, with associated KEGG ortholog (KO) gene identifiers. Because the genomes may be incomplete, it is possible that a KEGG module that is truly present in the organism will not be fully represented in the available data. MetaPathPredict first reconstructs both complete and incomplete KEGG metabolic modules, then predicts whether incomplete modules are in fact present. Input annotations can come from tools such as KofamScan (*Aramaki et al., 2020*), DRAM, blastKOALA (*Kanehisa et al., 2016*), ghostKOALA (*Kanehisa et al., 2016*), or a custom list of KO identifier gene annotations. MetaPathPredict classification

models produced accurate results on held-out test genome annotation datasets even when the data were highly incomplete. A set of two deep learning models.

(5 hidden layers each) made predictions with a high degree of precision on all test datasets and with high recall on genomes with an estimated completeness as low as 30%. One model was trained to classify the presence or absence of 96 KEGG modules that were present in ≥10% and≤90% of training genomes. The second model classifies 94 modules with an imbalanced profile of presence/ absence (i.e. were present in <10% or>90% of training genomes). False positive predictions were rare in all tests, while false negatives increased when predictions were made with highly incomplete gene annotation information, as would be expected. We believe that MetaPathPredict is a valuable tool to further enhance studies of metabolic potential in environmental microbiome studies as well as synthetic biology efforts.

## Results and discussion

MetaPathPredict is designed to predict the presence of a metabolic module even when annotation support for that module is incomplete, for example due to incomplete sequencing/annotation of the constituent proteins. It was trained on both complete and down-sampled genomes for this task. Complete genomes containing the genes required to non-redundantly complete a KEGG module were labelled as containing the module, otherwise the module was labelled as absent. To create down-sampled genomes for training, protein annotations were randomly removed form complete genomes in increasing increments while still retaining KEGG module class labels (from those complete genomes). To assess MetaPathPredict's efficacy in this 'gap filling' task, we performed a variety of benchmarking experiments in which the complete genomes/proteomes were down-sampled to artificially produce incomplete modules.

MetaPathPredict's exhibited superior performance to other recently developed metabolic pathway reconstruction and prediction approaches. Its performance metrics on held-out test datasets suggest its models predict with high fidelity when at least 30% of gene annotations are recovered from a reconstructed genome (*Figure 2*). The efficacy of MetaPathPredict models was assessed using incomplete gene annotation data simulated from whole genomes, as well as from genomes reconstructed from reads that had been randomly down-sampled. We further benchmarked MetaPathPredict against custom presence/absence classification rules, and existing gap-filling tools METABOLIC and Gapseq.

### Benchmarking MetaPathPredict on down-sampled NCBI RefSeq and GTDB data

We compared the performance of MetaPathPredict's deep learning models to two classes of competitor classifiers: naive rule-based classifiers and various other machine learning model architectures. The evaluation was performed on test datasets comprised of isolate and metagenome-assembled genomes from GTDB and NCBI (30,596 total genomes; see Methods). When evaluating with the same sets of randomly down-sampled gene annotations, we found that each competing method showed poorer performance than MetaPathPredict (*Figure 2*). We assessed two naïve classification methods. First, we devised a classification rule based on the completeness of a KEGG module relative to the number of genes retained after downsampling: if, in a down-sampled genome, the number of genes involved in a KEGG module are present in at least the same proportion of all genes retained, the KEGG module is classified as 'present,' otherwise it is labeled 'absent'. For example, if 50% of gene annotations were removed from a genome during downsampling, then any KEGG module for which 50% of its associated genes are retained would be reported as 'present'. The results of this naive approach (*Figure 2*) show that the relative completeness of a KEGG module alone is not a robust classification strategy. The second classification rule that we tested was: for all gene annotation sets in the dataset, if any genes were present in an annotation set that were unique to a KEGG module (relative to other modules) then the module was classified as 'present', otherwise it was labelled 'absent'. The results of this naive approach (*Figure 2*) suggest that the presence of rare protein annotations or genes unique to a certain KEGG module is not always a strong indicator of the presence of a module in a genome. Ultimately, the performance of these naive classifiers indicate that MetaPathPredict's models have the advantage of incorporating information from genes outside of KEGG modules. We additionally compared the performance of various machine learning model architectures. Of these,

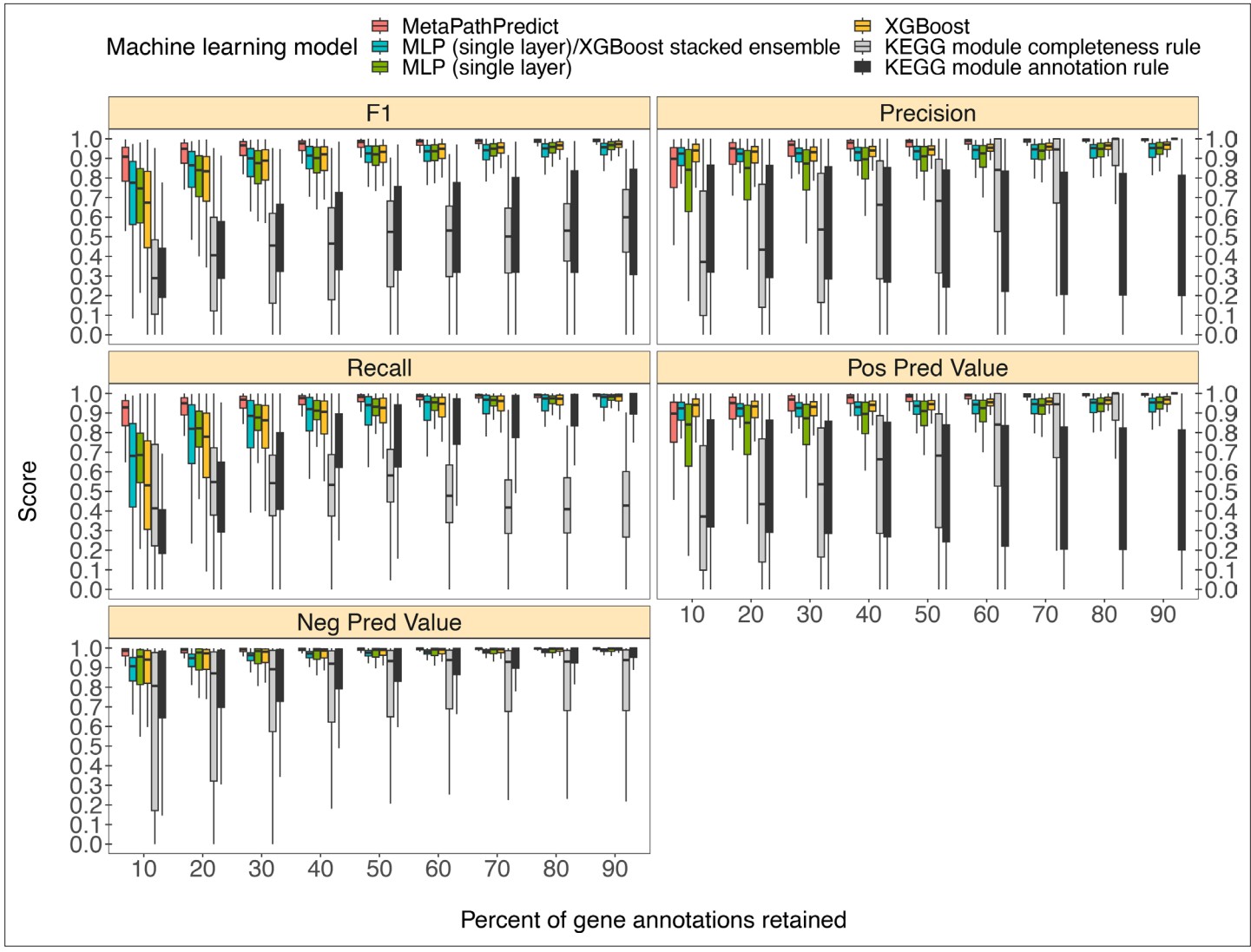

**Figure 2.** Comparison of performance metrics of MetaPathPredict's pair of deep learning multi-label classification models to next-best performing XGBoost, single-layer neural network, and XGBoost/single-layer neural network stacked ensemble machine learning models as well as two naïve classification rules. Down-sampled gene annotations of high-quality genomes used in held-out test sets are from NCBI RefSeq and GTDB. Each boxplot displays the distribution of model performance metrics for predictions on randomly sampled versions of the gene annotation test sets in downsampling increments of 10% (90% down to 10%, from right to left). The binary classifier performances are based on the classification of the presence or absence of KEGG modules in the complete versions of the gene annotations that were down-sampled for model testing.

The online version of this article includes the following source data for figure 2:

**Source data 1.** Column 'Module name' contains the shorthand identifiers from the KEGG database that correspond to KEGG modules; 'Percent of protein families retained' contains the percent of protein family presence/absence annotations retained during protein family downsampling; 'Model type' corresponds to the machine learning architecture or classification rule; 'Metric' lists the performance metric; 'Score' contains the value for the performance metric.

the XGBoost, neural network (single hidden layer) and XGBoost/neural network (single hidden layer) stacked ensemble architectures were the next-best performing models and are included in *Figure 2*.

MetaPathPredict's deep learning strategy produced the best observed performance. Mean F1 score (a summary metric of the predictive performance) of the models was 0.96 when predicting on test datasets in which 30–90% of gene annotations had been retained. MetaPathPredict rarely made false positive predictions based on data from highly incomplete gene annotation sets; the average precision of the models was consistently above 0.94 for all held-out test sets. MetaPathPredict also did not misclassify most negative class observations. The recall of MetaPathPredict's models was greater than 0.96 on average for test datasets containing at least 30% of the complete gene

annotation data. The mean recall decreased to 0.89 and the mean precision decreased to 0.86 on genomes containing only 20% or less of the complete gene annotation data. The models' ability to achieve notably high recall even with significantly reduced sampling rates implies that it compensates for limited sequence availability by becoming more assertive in labeling a module as 'present' at the cost of decreased precision.

## Benchmarking MetaPathPredict against Genomes from arth's Microbiomes repository MAGs

MetaPathPredict was further tested on gene annotations from a set of 40 high-quality metagenome-assembled genomes from the Genomes from Earth's Microbiomes (GEM; *Nayfach et al., 2021*) genome repository. This repository contains a set of MAGs recovered from a diverse array of environments that make it ideal for benchmarking MetaPathPredict's performance (*Figure 3*). The MAGs selected from the repository had an estimated completeness of 100 and estimated contamination of 0, MIMAG quality score of 'High Quality'. The genomes belonged to 7 taxonomic phyla and were recovered from 9 different environments, primarily from human-associated and built environment metagenomes (see *Appendix 1—figure 1* for GEM genome taxonomic distributions and environmental sources). We created a set of 9 GEM datasets by randomly downsampling the data to retain 10% to 90% of gene annotations (in 10% increments) as in the previous section. MetaPathPredict classified the presence/absence of KEGG modules in each MAG. Overall, results were comparable to MetaPathPredict's performance on the GTDB/NCBI benchmark. The models excelled at predicting the presence or absence of KEGG modules in genomes when at least 40% of gene annotations were randomly retained. Predictions were less reliable though still accurate when 30% or less gene annotation data was retained.

## Benchmarking MetaPathPredict against existing tools on a dataset with down-sampled reads

In addition to model assessments made through down-sampling protein annotations, we evaluated a second set of held-out test set genomes from the GTDB/NCBI dataset (n=50). In this analysis, the sequence reads for each genome were randomly down-sampled to simulate genomes incompletely recovered from an environmental sample. This analysis replicates situations with lower sequencing coverage, which can cause proteins to be unobserved due to incomplete or error-filled assemblies. As an example, using only 3% of reads (equivalent to an average of 1.5 x coverage of the genomes), roughly 86% of the genomes' proteins were assembled; meanwhile a reduction to 1% of reads caused assembly to recover ~40% of proteins. MetaPathPredict's performance on this test set resembled protein annotation random sampling results (*Figure 4a*, *Figure 4b*), although with greater loss in precision for down-sampling <3%.

MetaPathPredict had an average F1 score for all 190 modules of 0.96 on the second held-out test set observations that had an average estimated genome completeness of at least 30%. The similarity of these results to the gene annotation downsampling approach validates the latter approach that was used more broadly to assess MetaPathPredict.

In addition to evaluating MetaPathPredict against our custom competitor models, we tested the software METABOLIC, which is a command line tool that performs gene annotations and estimates the completeness of individual KEGG modules in genomes and prokaryotic microbial communities (*Figure 4a*). METABOLIC showed much poorer recall at all levels of read sampling.

MetaPathPredict was also compared to another gap filling tool, Gapseq (*Figure 5a*). Gapseq makes predictions of the presence or absence of KEGG pathways, and thus indirectly makes predictions of all modules and reactions they contain (instead of predictions for individual modules or reactions). We facilitated the direct comparison of MetaPathPredict to Gapseq by classifying a single KEGG pathway to be present if all modules it contained were predicted 'present' by MetaPathPredict. MetaPath-Predict outperformed predictions made by Gapseq, particularly on genomes with low read sampling prior to assembly.

## Analysis of model feature importance using SHAP

Though the neural networks of MetaPathPredict produce accurate predictions of module presence/absence, it is not immediately clear what input features contribute to its decision-making process. To

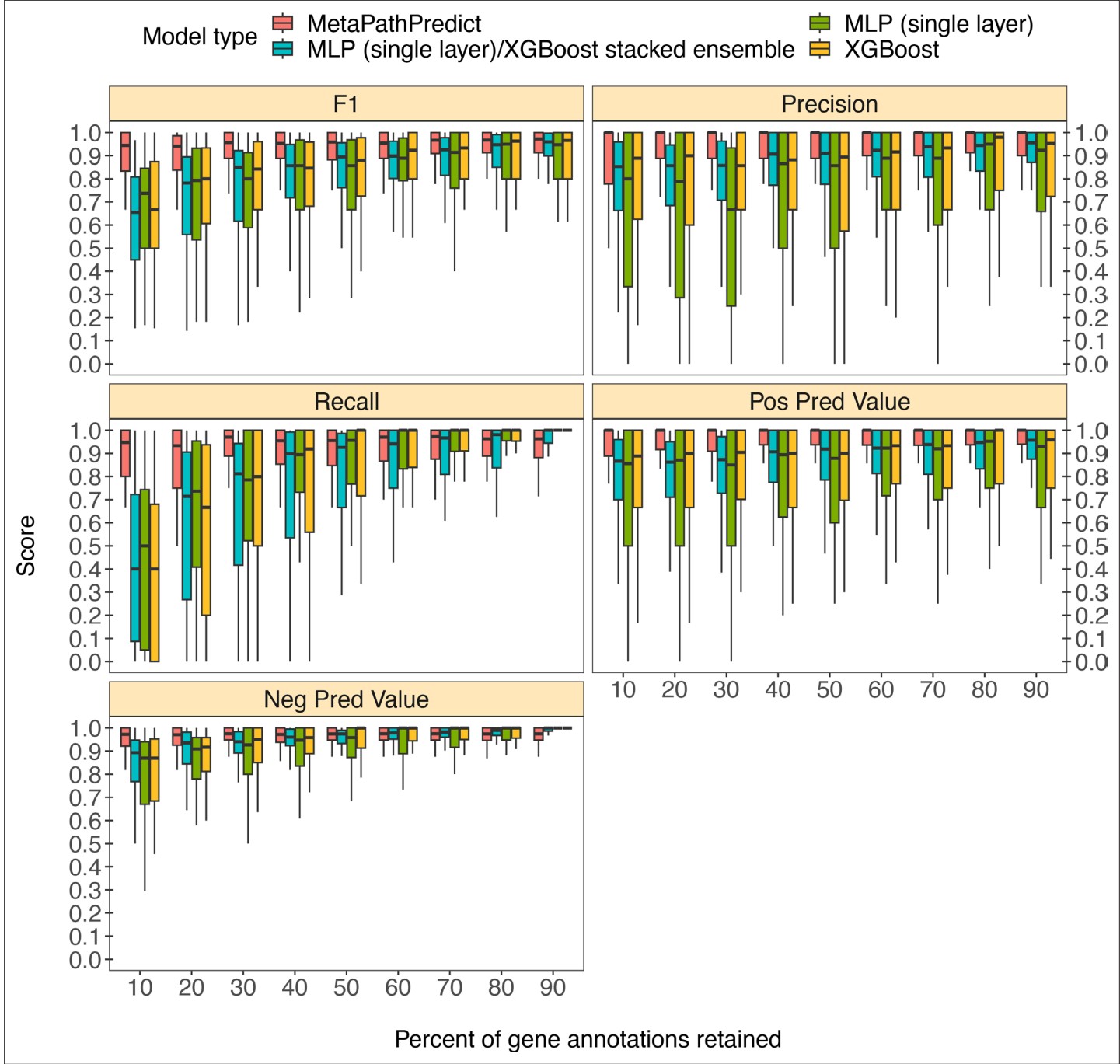

**Figure 3.** Boxplots of performance metrics of MetaPathPredict models on high-quality bacterial GEM MAGs (n=40). Model performance metrics are for predictions on down-sampled versions of GEM genome gene annotations in decreasing increments of 10% (retaining 10–90% of the annotations in each test set). MetaPathPredict's deep learning models were benchmarked against XGBoost and neural network model architectures.

The online version of this article includes the following source data for figure 3:

**Source data 1.** Column 'Module name' contains the shorthand identifiers from the KEGG database that correspond to KEGG modules; 'Percent of protein families retained' contains the percent of protein family presence/absence annotations retained during protein family downsampling; 'Model type' corresponds to the machine learning architecture; 'Metric' lists the performance metric; 'Score' contains the value for the performance metric.

gain some insight into this, we calculated the importance of the various features of MetaPathPredict's models using the SHAP method (*Lundberg and Lee, 2017*), a mathematical method to explain the predictions of machine learning models. Features with large absolute SHAP values play an important role in calculating a model's predictions. SHAP values in the first model (trained to classify modules

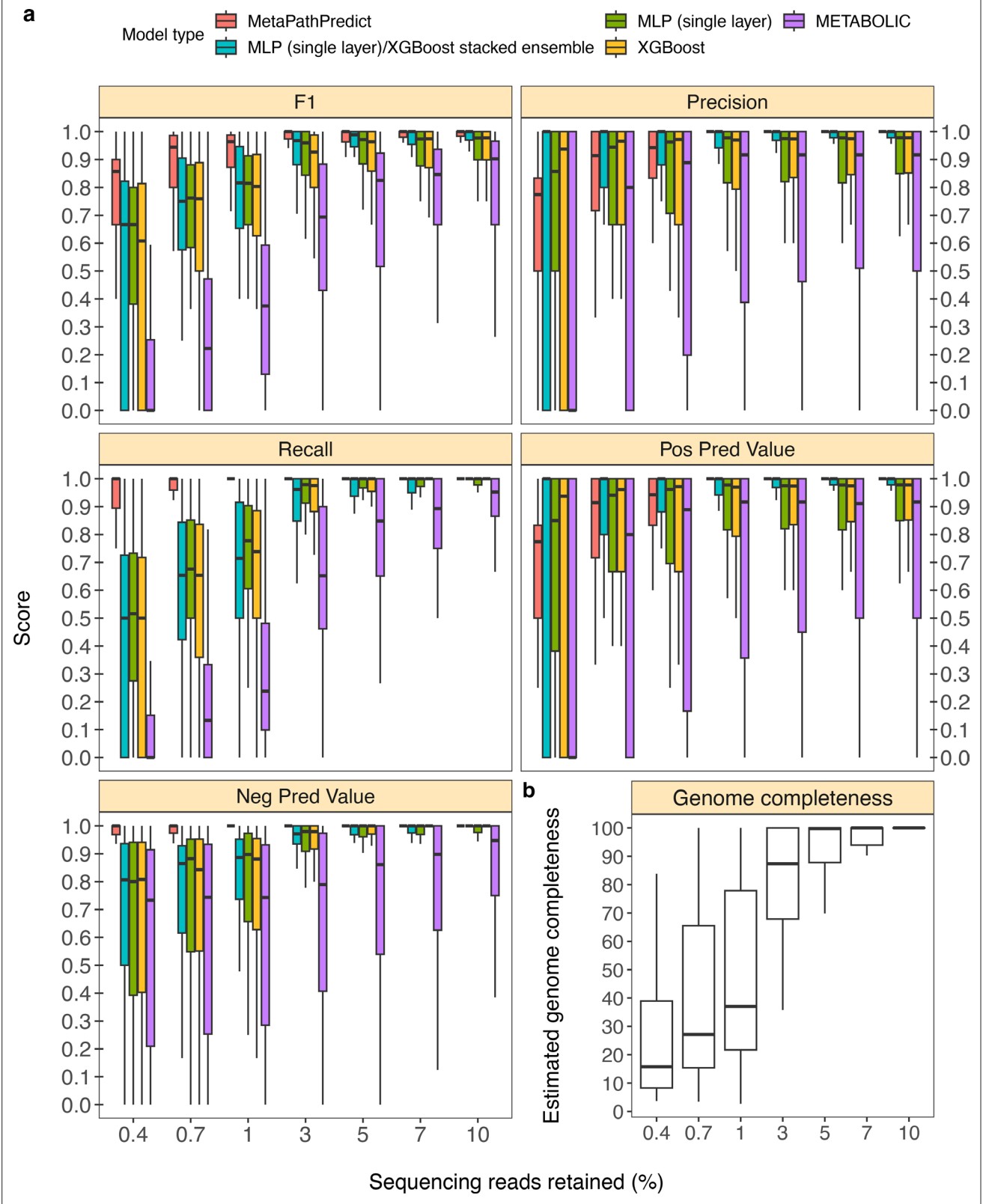

**Figure 4.** Performance metrics boxplots of two deep learning classification models. Down-sampled sequence reads of high-quality genomes used as a second held-out test set are from NCBI RefSeq and GTDB databases. (**Panel a**) Boxplots display the distribution of model performance metrics for predictions of KEGG module presence/absence on simulated incomplete genomes down-sampled at the sequence read level by MetaPathPredict models, various next-best performing machine learning architectures, and METABOLIC. Downsampling increments were chosen based on average

*Figure 4 continued on next page*

*Figure 4 continued*

estimated completeness of the test set genomes at each increment to reflect a range of estimated completeness thresholds. (**Panel b**) Average estimated genome completeness distributions of test set genomes that were down-sampled at the sequence read level using SeqTK and then assembled with SPAdes.

The online version of this article includes the following source data for figure 4:

**Source data 1.** Column 'Module name' contains the shorthand identifiers from the KEGG database that correspond to KEGG modules; 'Percent of sequencing reads retained' contains the percent of sequencing reads retained during sequencing read downsampling; 'Model type' corresponds to the machine learning architecture or annotation tool; 'Metric' lists the performance metric; 'core' contains the value for the performance metric.

present in ≥10% and≤90% of training genomes) indicated that 30 of the top 100 most important features (genes) influencing predictions were direct components of KEGG modules predicted by this model (**Supplementary file 1b**). In the second model (classifies modules present in <10% or>90% of training genomes), 37 of the 100 most influential features were part of KEGG modules this model was trained to predict.

The two models share 14 most important features out of the top 50 that are not part of KEGG modules. We examined these top features and found that there were a number of proteins annotated as sensors or transcriptional regulators (**Supplementary file 1b**). Also, we noted a number of transporters annotated as top features in both models and, interestingly, factors related to pathogenesis like toxins and mobile elements. Given the multi-label architecture of our models, it is difficult to draw direct conclusions from SHAP analysis. However, it is clear that MetaPathPredict's predictions are in part influenced by select genes present in KEGG modules, and also to a larger extent by genes not directly participating in KEGG module reactions.

## Conclusion

MetaPathPredict is an open-source tool that can be used to characterize the functional potential of one or more sample genomes by detecting complete KEGG modules and predicting the presence or absence of those that are incomplete or missing. The tool accepts sets of gene annotations of individual genomes in KO gene identifier format as input. This type of annotation format can be acquired by annotating a genome of interest using KEGG-based annotation tools such as KofamScan (**Aramaki et al., 2020**), DRAM, blastKOALA (**Kanehisa et al., 2016**), or ghostKOALA (**Kanehisa et al., 2016**). MetaPathPredict also provides gene gap-filling options by listing putative KO gene annotations that could fill in missing gaps in predicted modules.

MetaPathPredict further validates the use of gene family presence or absence within a genome as a feature for bacterial metabolic function predictions. The performance metrics of MetaPathPredict on NCBI/GTDB and GEM test datasets validated the use of deep learning models to predict the presence/absence of KEGG metabolic modules with high fidelity on sparse to near-complete bacterial genomes. MetaPathPredict's multi-label classification models consistently made predictions with high precision and recall on simulated and real genomes using gene annotation and sequence read downsampling methods. The predictive power of the deep learning models was most limited when predicting on 10%–30% of protein annotations, and when the mean estimated completeness of reconstructed genomes from down-sampled reads was below 30%. We suggest that optimal performance with MetaPathPredict can be achieved when at least 40% of a genome has been recovered in an input bacterial gene annotation dataset.

Based on our performance tests of MetaPathPredict, the recall of its models was robust (mean >0.9) even when protein sets were down-sampled to 10%. However, MetaPathPredict also surprisingly shows a decrease in precision (i.e. an increase in false calls of module presence). This, combined with surprisingly high recall rates at such low sampling rates, suggests that the model directly compensates for low general sequence availability by increasing aggressiveness in calling a module 'present'. This over-exuberant positive class prediction problem arose in our analyses only when <30% of gene annotation data was retained. Though such low sampling rates are not expected to be typical, it suggests an opportunity for method improvement.

Due to the multi-label architecture of MetaPathPredict's models, it is difficult to draw connections between the important features identified for the models and individual KEGG modules. However, the presence of sensing proteins (e.g. iron sensing and chemotaxis), pathogen proteins (e.g. toxins and lysins), and transporters in these lists may indicate the contribution of lifestyle and environmental

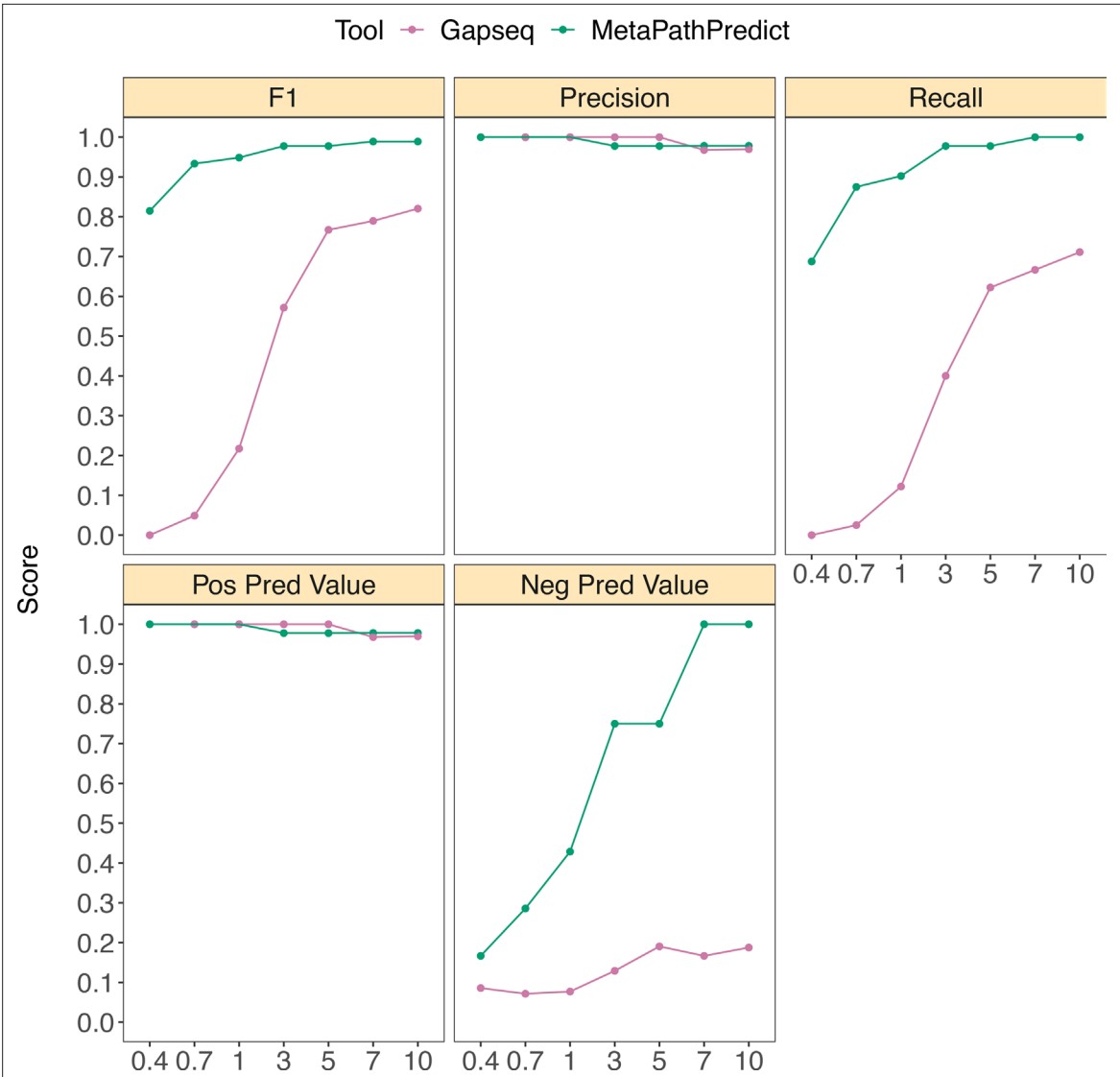

**Figure 5.** Performance metrics boxplots of MetaPathPredict and Gapseq predictions for KEGG pathway map00290 (Valine, leucine, and isoleucine biosynthesis) which contains KEGG modules M00019, M00432, M00535, and M00570. For MetaPathPredict predictions, the whole KEGG pathway was considered present if the aforementioned KEGG modules were all present. Down-sampled sequence reads of high-quality genomes used as a second held-out test set are from NCBI RefSeq and GTDB databases. Line segments display model performance metrics for MetaPathPredict and Gapseq predictions of KEGG pathway map00290 presence/absence on simulated incomplete genomes down-sampled at the sequence read level. Downsampling increments were chosen based on average estimated completeness of the test set genomes at each increment to reflect a range of estimated completeness thresholds.

The online version of this article includes the following source data for figure 5:

**Source data 1.** 'Percent of sequencing reads retained' contains the percent of sequencing reads retained during sequencing read downsampling; 'Model type' corresponds to the machine learning architecture or annotation tool; 'Metric' lists the performance metric; 'Score' contains the value for the performance metric.

factors in predicting presence or absence of individual modules. Additionally, transcriptional regulators may be important due to their outsized influence on the expression of many genes (and thus modules) in each organism. Perhaps the most intriguing finding was that components of mobile elements (transposases) were found to be important features of both models. This could indicate that insertional elements are being used by the model to indicate, for example, evolutionary lineage, which could be used to inform predictions of KEGG module composition.

MetaPathPredict facilitates more complete and accurate reconstruction of the metabolic potential encoded within bacterial genomes from a diverse array of environments and will enhance the ability to infer what metabolisms they are capable of, and/or how they may respond to perturbations. MetaPathPredict connects the field of machine learning with the growing community of environmental microbiologists using genomic sequencing techniques and will help transform and improve the way they work with environmental genomic datasets.

## Materials and methods
### Filtering genome database metadata, downloading high-quality genomes, and gene annotations

The NCBI RefSeq (Release 205) database metadata file was downloaded and filtered to retain only the information for all bacterial genomes classified as 'Complete genome'. These are defined on the NCBI assembly help webpage: 'all chromosomes are gapless and have no runs of 10 or more ambiguous bases (Ns), there are no unplaced or unlocalized scaffolds, and all the expected chromosomes are present (i.e. the assembly is not noted as having partial genome representation). Plasmids and organelles may or may not be included in the assembly but if present, then the sequences are gapless'. This resulted in 17,491 complete NCBI genomes.

The GTDB bacterial metadata file for release 95 was downloaded and filtered to keep the information for all genomes with an estimated completeness greater than 99, an estimated contamination of 0, and a MIMAG (*Bowers et al., 2017*) quality score of "High Quality". A total of 30,760 non-redundant bacterial genomes from the two database metadata files were downloaded using the ncbi-genome-download command line tool (*Blin, 2023*). The RefSeq genomes (17,491 total) were downloaded from the RefSeq FTP server, and the GTDB genomes (13,105 total) were downloaded from the GenBank FTP server (*Appendix 1—figure 2*). Genes were called using Prodigal (*Hyatt et al., 2010*), and the KofamScan command line tool (*Aramaki et al., 2020*) was used to generate gene annotations (in KO gene identifier format) for all of the genomes using the KOfam set of HMMs available for download from the KEGG database (*Kanehisa, 2002*). KofamScan-derived annotations had to surpass their HMM's adaptive scoring threshold to be included in the training dataset. This approach provides resilience to using specific e-value cutoffs by preventing inflation of our training and assessment datasets with less-confident gene annotations.

### Formatting gene annotation data, fitting KEGG module classification models

The full dataset of simulated incomplete genomes (n=305,960) was split so that 75% of genomes were used for training and the remaining 25% as a test dataset. The training dataset was further split into 80% training/20% validation test sets. Each observation in the train/test/validation datasets contained a vector of length 8,853 that consisted of KO gene identifier (protein family) presence/absence indicated by ones and zeroes, respectively.

Training and test sets contained both complete and incomplete gene annotations of bacterial genomes from a diverse array of phyla (*Appendix 1—figure 2*). The incomplete annotations used in training and testing of MetaPathPredict's models were constructed from complete genome annotation observations that were randomly down-sampled to retain 10–90% of the total gene content while the presence/absence class labels were kept unchanged for all down-sampled data. All complete and down-sampled versions of genomes were retained. The training datasets had a size of 305,960 observations, and the test datasets each contained 76,490 observations. The percent of observations with a positive class (a complete KEGG module 'present' in the gene annotations) in the training and test datasets varied, with a mean of 26.2% (*Appendix 1—figure 3*).

The gene copy number data of the downloaded genomes was formatted in a matrix containing KO gene identifier presence/absence (1 or 0, respectively) in columns and genomes in rows. The label of each model was the presence/absence (1 or 0) of a KEGG module, as was determined using the KEGG modules downloaded from the KEGG database and the Anvi'o Python module (*Eren et al., 2015*). The 'unroll_module_definition' function from the Anvi'o module was utilized with downloaded KEGG module data to create a list of all possible KEGG Ortholog combinations to complete each module. For the module to be categorized as present, at least one possible combination of every step of the

module had to be present in a genome, otherwise it was designated as absent. Two models were constructed for 190 KEGG modules for which at least 306 (0.1%) of the complete genomes (n=30,596) contained the module genes, due to an improvement in performance when two models were trained (one model for the more balanced labels, one for highly imbalanced labels) instead of one. The models were trained using the gene annotation data of the genomes consolidated from the NCBI and GTDB databases. The first model was trained to classify modules within ≥10% and≤90% of training genomes, while the second model classified modules within <10% or>90% of training genomes. The constructed models classify the presence or absence of complete KEGG modules based on the gene annotations of a genome.

A deep learning classification approach was chosen to model the relationship between whole genome KO gene identifier annotation data and the presence of metabolic modules. The same training data was used to train both of the models. MetaPathPredict is built on the Keras deep learning library (*Sattler et al., 2015*). L2-regularization was utilized to adjust hidden unit weights during training, with a learning rate of 0.001. Features used in each training dataset for classification were the presence or absence of protein families that were assigned KO gene identifiers. A deep learning architecture consisting of one input layer, five hidden layers, and one output layer were used as the machine learning architectures in MetaPathPredict's models. The input layer consisted of the presence/absence vector of KO gene identifiers (n=8,853), and the hidden layers each contained 2048 hidden units and were fully connected. The output layer of the first and second models contained 94 and 96 nodes for a total of 190 module presence/absence predictions when prediction outputs from both are combined.

Stratified sampling is a sampling method that ensures that all groups within the training and test data are represented in the same proportion as they are in the population as a whole. A multi-label stratified sampling method (*Sechidis et al., 2011*) was used to generate 75% train/25% test dataset splits that each contained data observations with preserved proportions of positive ('KEGG module present') and negative ('KEGG module absent') classes that were present in the genome dataset (see boxplot of the distribution of module presence/absence classes in *Appendix 1—figure 2*, and an example of a held-out test dataset in *Appendix 1—figure 4*). The training dataset was further separated into 80% train/20% validation dataset splits to fit the deep learning models.

The binary cross entropy loss function was used in tandem with the Adaptive Moment Estimation (Adam) optimizer. The input and hidden layers utilized the rectified linear unit (ReLu) activation function; the output layer contained a sigmoid activation function. Dropout (*Srivastava, 2014*) was applied to 10% of edges at all layers except the final layer to avoid overfitting the training data. The input and hidden layers utilized the 'he_uniform' layer weight initializer, and each of these layers contained 2,048 hidden units.

We assessed and benchmarked MetaPathPredict's models against two naive classification methods. First, we devised a simple model that predicted the presence of a KEGG module if, after downsampling test sets of gene annotations, the proportion of module genes present in the dataset was greater than or equal to the percentage of annotations retained in the dataset. If the proportion of genes involved in a KEGG module were present in a dataset observation at least equivalently to the proportion of gene annotations retained after downsampling, the module was classified as 'present', otherwise it was classified as 'absent'. The second naïve classification rule was: for all gene annotation sets in the dataset, if any genes were present in an annotation set that were unique to a KEGG module (relative to all other KEGG modules) then the module was classified as 'present', otherwise it was labeled 'absent'. We additionally benchmarked MetaPathPredict's deep learning models against several other machine learning model types including single-layer neural network, XGBoost, and neural network/XGBoost stacked ensemble models, each trained on the same input data.

## Evaluating models on test data, including test data randomly downsampled to simulate varying degrees of genome incompleteness

Each of MetaPathPredict's models was validated on a held-out test set consisting of a combination of 76,490 complete and simulated incomplete genomes, and the performance metrics were extracted using the Scikit-learn (*Pedregosa, 2011*) Python module. The genome annotations in each test set were created by randomly downsampling complete genomes to simulate recovered gene annotations from incomplete genomes. 10% to 90% of genes from each annotation set were randomly retained

**Table 1.** Definitions of machine learning model performance metrics used to assess MetaPathPredict models.

| Metric | Definition |
|---|---|
| Precision | true positive/(true positive +false positive) |
| Recall | true positive/(true positive +false negative) |
| Specificity* | true negative/(true negative +false positive) |
| F1 score | 2 × ((precision ×recall)/(precision +recall)) |
| Positive predictive value | recall ×prevalence / (recall ×prevalence) + (1 – specificity) × (1 – prevalence) |
| Negative predictive value | specificity × (1 – prevalence) / ((1 – recall)×prevalence) + (specificity × (1 – prevalence)) |
| Prevalence* | (true positive +false negative) / (true positive +false positive +true negative +false negative) |

*Specificity and prevalence are defined due to their use in the definitions of negative and positive predictive value.

(in increments of 10%) and used as input for MetaPathPredict predictions of KEGG module presence/ absence. The performance metrics used in evaluating the models were precision, recall, F1 score, positive predictive value, and negative predictive value (*Table 1*).

## Testing models with a set of high-quality metagenome-assembled genomes from the Genomes from Earth's Microbiomes online repository

MetaPathPredict was further validated on another test set of genome annotations extracted from the GEM repository of MAGs. The GEM metadata file was downloaded from the repository and filtered to retain a random sample of 40 MAGs with a CheckM2 (*Chklovski et al., 2023*) estimated completeness of 100, an estimated contamination of 0, and a MIMAG quality score of 'High Quality'. The method for this assessment was the same as was described above for testing MetaPathPredict model performances on the held-out test data.

## Evaluating models on test data down-sampled at the read level

A second held-out set of complete genomes (n=50), independent of the training dataset, was downloaded from NCBI/GTDB databases using the *SRA, 1988* and SRA explorer (*Phil Ewels, 2024*). The raw sequencing reads were filtered using fastp (*Chen et al., 2018*), and the quality-filtered reads were randomly down-sampled using seqtk (*Li, 2023*). Down-sampled reads were assembled into genomes using the SPAdes assembler (*Bankevich et al., 2012*), genes were called with Prodigal and then annotated using KofamScan. MetaPathPredict's deep learning models were then used to predict the presence or absence of all 190 KEGG modules in each genome and predictions were then cross-referenced with their known presence/absence based on the unmodified test dataset. In addition to simple approaches described above, the METABOLIC (*Zhou et al., 2022*) and Gapseq (*Zimmermann et al., 2021*) tools were evaluated on the same benchmark dataset. Both tools were used with default settings. Gapseq makes predictions of the presence or absence of entire KEGG pathways, and therefore it was benchmarked against MetaPathPredict by evaluating predictions for the presence or absence of the KEGG pathway map00290 (Valine, leucine, and isoleucine biosynthesis). This pathway consists of KEGG modules M00019, M00432, M00535, and M00570. In order to facilitate a direct comparison to Gapseq's predictions, the whole KEGG pathway was considered present if the aforementioned KEGG modules were all predicted as present by MetaPathPredict, otherwise it was classified as absent.

### Gapfilling for incomplete modules predicted as present

MetaPathPredict provides enzyme gap-filling options for KEGG modules predicted as present by suggesting putative KO gene annotations missing from an input genome's gene annotations that could fill in missing gaps in predicted modules.

## Acknowledgements

Geller-McGrath acknowledges funding from the Department of Energy (DOE) SCGSR Fellowship for the 2020 Solicitation 2 in Computational Biology and Bioinformatics. We would like to thank A Solow (WHOI) for helpful initial discussion about statistical approaches. JWR and TJW were supported by NIH NIGMS R01GM132600. JEM, JWR, and TJW were supported by the DOE Office of Biological and Environmental Research (BER) through the "Machine-Learning Approaches for Integrating Multi-Omics Data to Expand Microbiome Annotation" project. PNNL is operated for the DOE by Battelle Memorial Institute under Contract DE-AC05-76RL01830.

## Additional information

### Competing interests

Kishori M Konwar: affiliated with Luit Consulting. The author has no financial interests to declare. The other authors declare that no competing interests exist.

### Funding

| Funder | Grant reference number | Author |
| --- | --- | --- |
| Department of Energy | SCGSR Program 2020 Solicitation 2 in Computational Biology and Bioinformatics | David Geller-McGrath |
| National Institutes of Health | NIGMS R01GM132600 | Jason E McDermott Travis J Wheeler |
| Department of Energy Office of Biological and Environmental Research | Machine-Learning Approaches for Integrating Multi-Omics Data to Expand Microbiome Annotation | Jason E McDermott Jack W Roddy Travis J Wheeler |

The funders had no role in study design, data collection and interpretation, or the decision to submit the work for publication.

### Author contributions

David Geller-McGrath, Conceptualization, Data curation, Software, Formal analysis, Validation, Investigation, Visualization, Methodology, Writing – original draft; Kishori M Konwar, Conceptualization, Supervision, Methodology, Writing – review and editing; Virginia P Edgcomb, Jack W Roddy, Travis J Wheeler, Jason E McDermott, Conceptualization, Methodology, Supervision, Writing – review and editing; Maria Pachiadaki, Conceptualization, Supervision, Writing – review and editing

### Author ORCIDs

David Geller-McGrath ⓘD https://orcid.org/0000-0002-5629-9983
Jason E McDermott ⓘD https://orcid.org/0000-0003-2961-2572

### Decision letter and Author response

Decision letter https://doi.org/10.7554/eLife.85749.sa1
Author response https://doi.org/10.7554/eLife.85749.sa2

## Additional files

### Supplementary files

• Supplementary file 1. Metadata for KEGG modules available within MetaPathPredict, mean SHAP values for both of MetaPathPredict's models, and training genome metadata. (**a**) MetaPathPredict KEGG Module information. The column "Module name" contains the name of all KEGG Modules that MetaPathPredict is trained to predict; "Module number" contains the module identifier; "Module class" contains module metadata including which group of KEGG metabolism the module belongs to. (**b**) SHAP results for the features in MetaPathPredict's models. The column "K number (Model 1)" contains the KEGG Ortholog gene identifier for each feature in Model 1. "Mean SHAP value (Model 1)" corresponds to the mean SHAP value for features in Model 1; highest mean SHAP values are listed at the top of the column in descending order. "KEGG Module (Model 1)" shows which KEGG module(s), if any, the features of Model 1 are present in; "Gene definition (Model 1)" contains gene annotation information for each feature in Model 1. The same column definitions are repeated for Model 2 and correspond to the same information as for Model 1. (**c**) Training genome metadata. The column "Genome ID" lists the names of genomes downloaded from the NCBI and GTDB databases; "Database" lists which database each genome was downloaded from (NCBI or GTDB); columns "Phylum", "Class", "Order", "Family", "Genus", and "Species" contain the associated taxonomic information for each genome.

• MDAR checklist

### Data availability

Genomic data used for creation of MetaPathPredict models is available from the NCBI Bacterial RefSeq Genomes database (https://ftp.ncbi.nlm.nih.gov/genomes/refseq/bacteria/, version 209) and the Genome Taxonomy Database (https://data.gtdb.ecogenomic.org/releases/latest/, version r95). The GEM genomes used for model benchmarking are available at the GEM repository (https://portal.nersc.gov/GEM/genomes/). The sequencing reads used for model benchmarking are available at the NCBI Sequence Read Archive website (https://www.ncbi.nlm.nih.gov/sra). The scripts used for all data processing, model training, model benchmarking, and figure creation used in this study are available in the following GitHub repository: https://github.com/Microbiaki-Lab/MetaPathPredict_workflow (copy archived at *McGrath, 2024*). The MetaPathPredict Python module is available from the following GitHub repository: https://github.com/d-mcgrath/MetaPathPredict and XetHub repository: https://xethub.com/dgellermcgrath/MetaPathPredict.

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

## Appendix 1

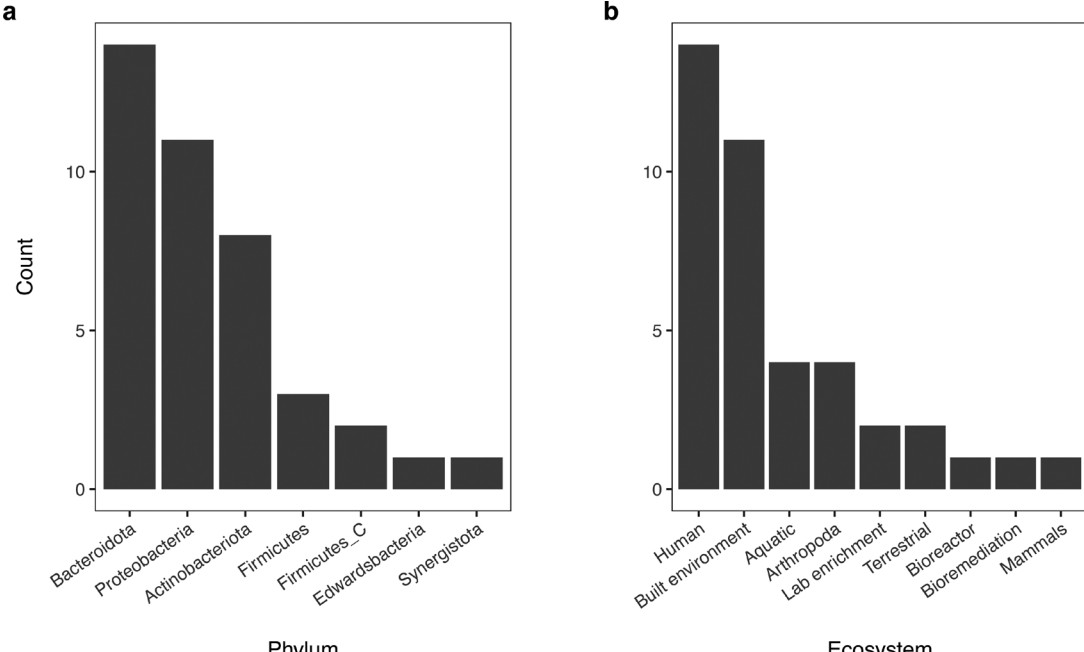

**Appendix 1—figure 1.** Distribution of phyla of bacterial genomes from the GEM repository used during model validation and associated environments they were recovered from. (Panel **a**) Bar chart of the taxonomic distribution of genomes (n = 40) from the GEM repository used during model validation. (Panel **b**) Bar chart of the environmental sources of metagenomes the MAGs from this test set were recovered from.

The online version of this article includes the following source data for appendix 1—figure 1:

**Appendix 1—figure 1—source data 1.** The column "Genome ID" lists the names of genomes downloaded from the GEM database; columns "Phylum", "Class", "Order", "Family", "Genus", and "Species" contain the associated taxonomic information for each genome; "Ecosystem category" lists the environment the genome was recovered from.

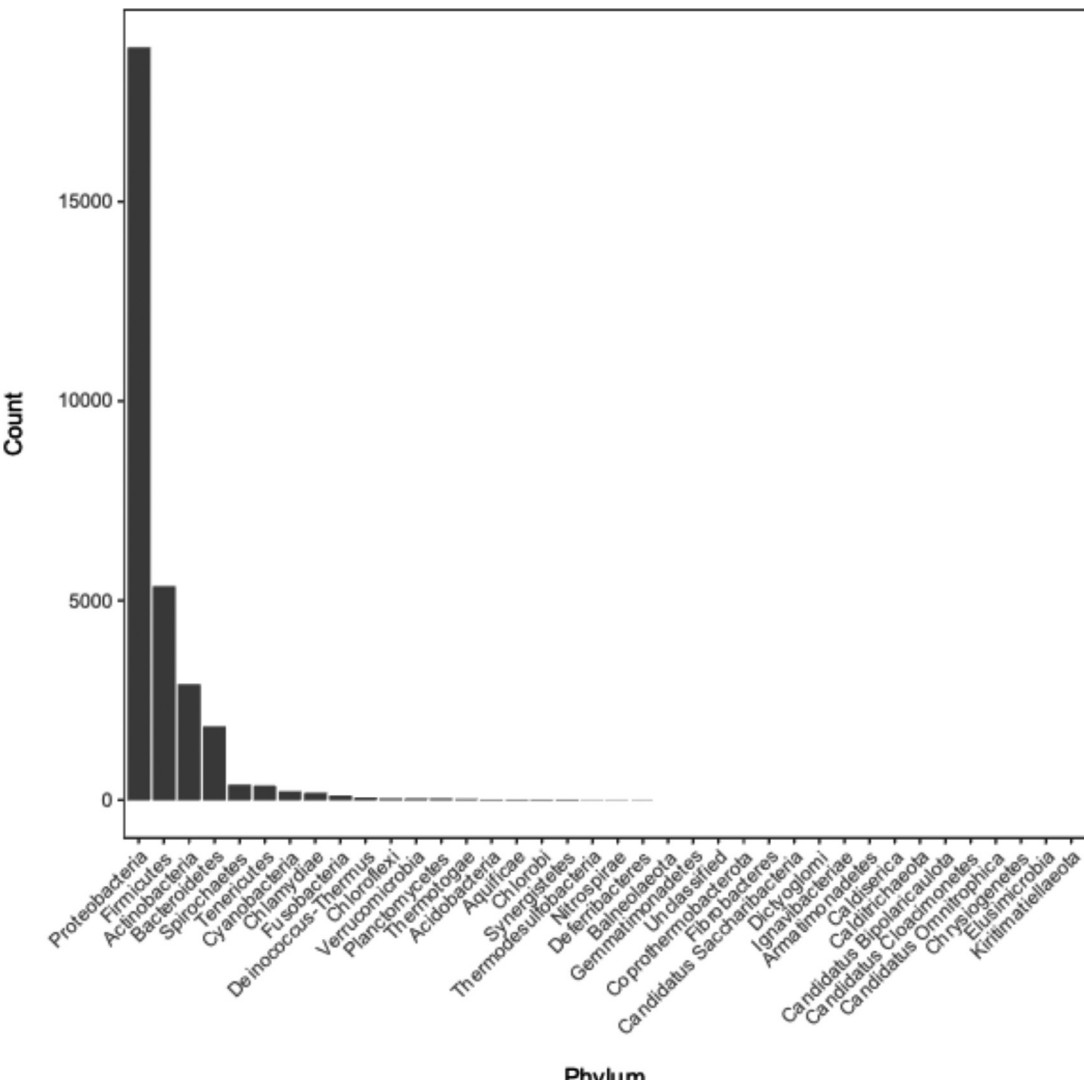

**Appendix 1—figure 2.** Distribution of phyla of bacterial genomes from which annotation data was used during model training and testing. See *Supplementary file 1C* for the full metadata table.

The online version of this article includes the following source data for appendix 1—figure 2:

**Appendix 1—figure 2—source data 1.** The column "Genome ID" lists the names of genomes downloaded from the NCBI and GTDB databases; "Database" lists which database each genomewas downloaded from (NCBI or GTDB); columns "Phylum", "Class", "Order", "Family", "Genus", and "Species" contain the associated taxonomic information for each genome.

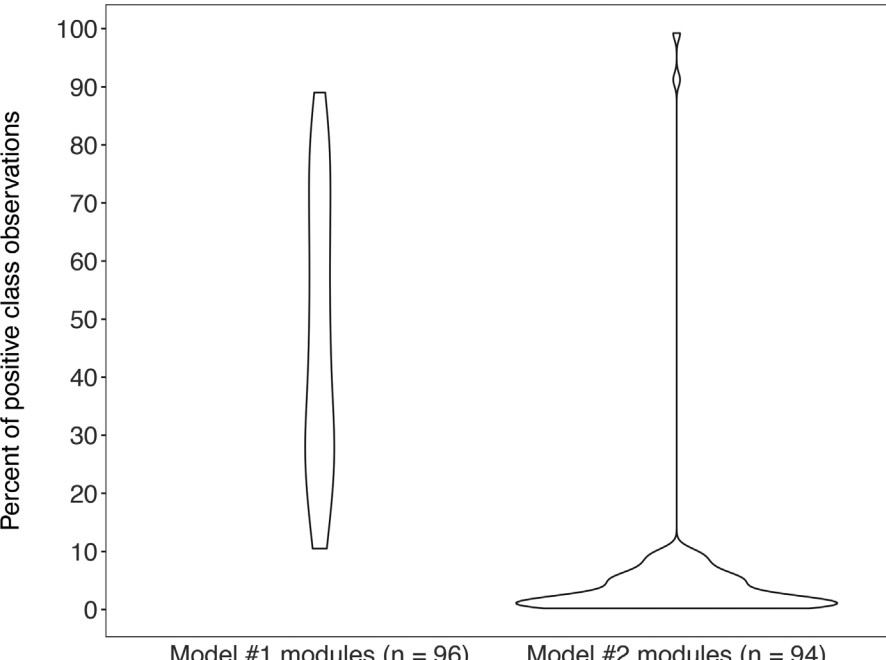

**Appendix 1—figure 3.** Violin plots of the percent of positive "KEGG module present" classes for genomes from MetaPathPredict's deep learning training and test datasets for both of its models (model #1 on the left-hand side; model #2 on the right-hand side). Each train/test split contains the same distribution of positive and negative classes.

The online version of this article includes the following source data for appendix 1—figure 3:

**Appendix 1—figure 3—source data 1.** The column "Module name" contains the name of all KEGG Modules that MetaPathPredict is trained to predict; "Prevalence" lists the percent of training genomes the complete module was detected in; "Model" lists the model the prevalence data corresponds to (Model 1/Model 2).

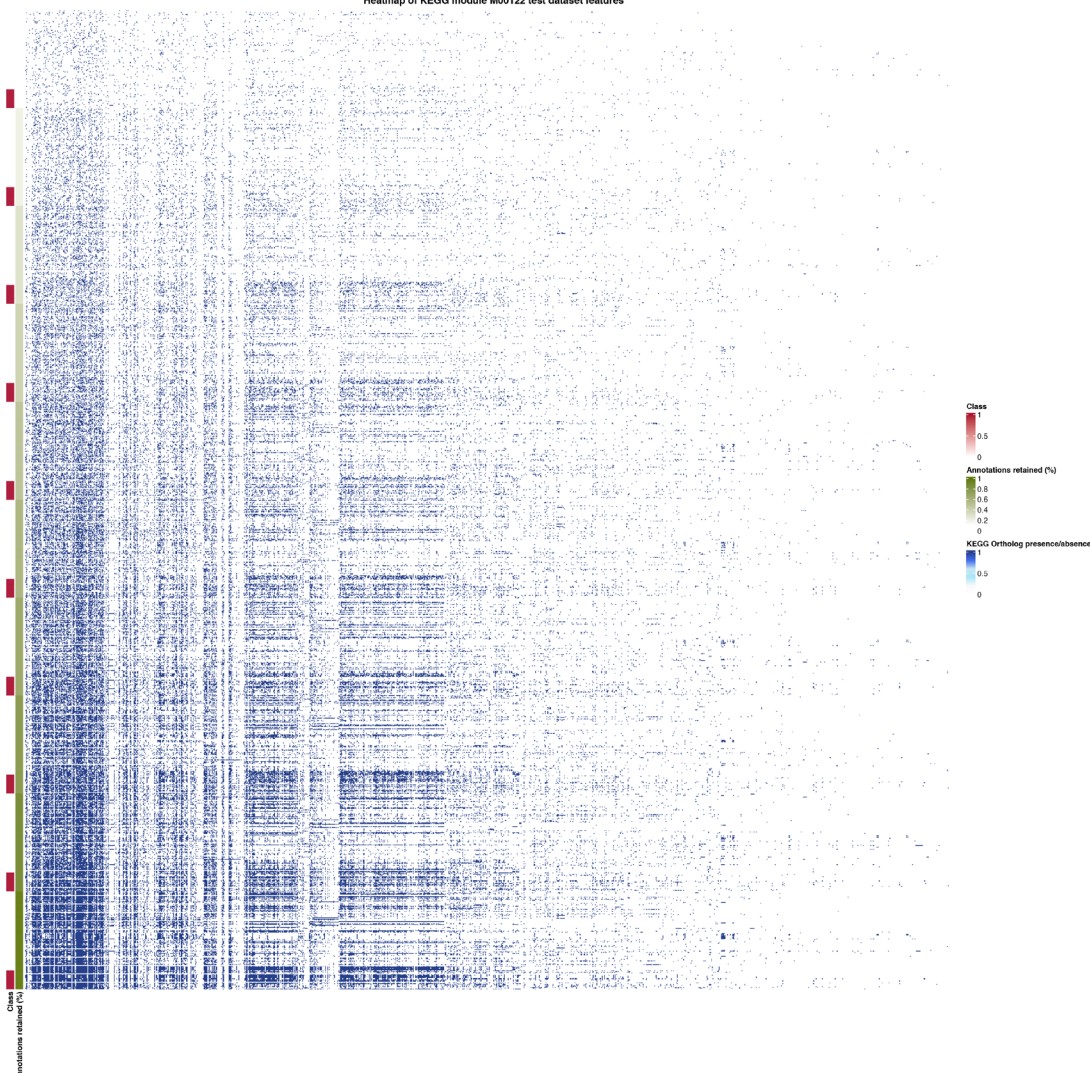

**Appendix 1—figure 4.** Heatmap of held-out test data for the set of features (KEGG Ortholog presence/absence) used by MetaPathPredict's deep learning models. The annotation row on the left-hand side of the plot is annotated with classes and predictions for KEGG module M00122 (cobalamin biosynthesis), and is sorted by the percentage of protein annotations retained in each observation (increasing in protein annotations retained from top to bottom).

The online version of this article includes the following source data for appendix 1—figure 4:

**Appendix 1—figure 4—source data 1.** The column "Module M00122" contains the class labels for the presence or absence of KEGG Module M00122 in the training dataset; "Proportion of protein families retained" contains the proportion of protein family presence/absence annotations retained during protein family downsampling; the remain columns correspond to the presence or absence of protein families in the training genomes.

