## [Editor Report]

This landmark study presents MetaPathPredict, a method that uses deep neural networks to predict the presence or absence of KEGG modules based on annotated features in the genome. The evidence supporting the conclusions is compelling, with a tool that allows for the prediction of KEGG modules in sparse gene sequence datasets.

---

## [Decision Letter]

**Decision letter after peer review:**

Thank you for submitting your article "MetaPathPredict: A machine learning-based tool for predicting metabolic modules in incomplete bacterial genomes" for consideration by *eLife*. Your article has been reviewed by 2 peer reviewers, and the evaluation has been overseen by a Reviewing Editor and Bavesh Kana as the Senior Editor.

Essential revisions (for the authors):

*Reviewer #1 (Recommendations for the authors):*

I have only a few questions and comments about the paper:

1. It's possible the training set itself will have some incomplete modules dues to a mixture of novel gene families, poor gene calls, or annotation error. What steps were taken to address this possibility? Were "nearly complete" modules considered "complete", and what was the threshold?

2. The "KofamScan command line tool" was used to annotate the training set for this method. Of course, other annotation methods (e.g. DRAM) may render significant differences in the resulting functional annotations. Did the authors test if the KofamScan-trained classifiers show similar accuracy/performance when applied to annotations from DRAM or some other competing tool?

3. Would it be possible to train the method on KEGG reaction IDs so other approaches that annotate reactions directly could be applied to the method? Not suggesting the authors do this work, but it might be worth mentioning in the discussion.

4. Did the authors look at the features chosen by the classifiers for various modules? It would be interesting to know how often the top-ranking features lie within the module or outside, and when outside, what kind of feature is used?

5. The link in the PDF to the git repo (https://github.com/d-mcgrath/MetaPathPredict) is corrupted somehow and doesn't work. I suspect the "-". Manually entering the address does get me to the repo.

*Reviewer #2 (Recommendations for the authors):*

Observation 1: Figures 2-5 present compelling evidence of MetaPathPredict's predictive capabilities. However, the authors have not discussed the potential mechanisms that could be discovered using the stacked ensemble of neural networks. In lines 171-173, the authors mentioned that MetaPathPredict's models incorporate information from genes outside of KEGG modules, but they have not elaborated on how such information can be interpreted. While predictive power is a commendable goal, machine learning models usually trade off explanatory power for predictive power. Therefore, the authors should discuss whether the stacked ensemble of neural networks could provide biological insights (https://doi.org/10.1038/s42256-019-0048-x).

Observation 2: The authors have primarily focused on predicting the presence of complete KEGG modules, which may result in predictions that are overly conservative regarding gene essentiality. For example, some KEGG modules may have functionally redundant reactions, and an incomplete KEGG module could still lead to a viable set of metabolic reactions. To address this issue, the authors could predict gene essentiality, carbon source, or metabolic product. (For further information, see https://doi.org/10.1186/s13059-021-02295-1.)

---

## [Author Response]

Essential revisions (for the authors):Reviewer #1 (Recommendations for the authors):I have only a few questions and comments about the paper:1. It's possible the training set itself will have some incomplete modules dues to a mixture of novel gene families, poor gene calls, or annotation error. What steps were taken to address this possibility? Were "nearly complete" modules considered "complete", and what was the threshold?

Thank you for identifying issues that may have arisen in constructing our training data. In order to classify an organism as containing a module, all steps in the module must have been present. We did not classify organisms containing, for example, 9 out of 10 of the steps in a module as containing the entire module. It is non-trivial to determine how many organisms contain a certain percent of a module given the varying number of steps that comprise each KEGG module. There are likely instances in which organisms with unannotated portions of their genomes do contain the missing steps of modules and were incorrectly labelled in the “absent” class. Though it would be nice for training data to be perfectly labeled, it is common for some level of noise (mis-labeled data) to be admitted into the training set, and for machine learning models to work in spite of this. However, we accounted for this issue by using a large number of genomes in our overall dataset: with training and test data generated from over 30,000 high quality isolate genomes and metagenome-assembled genomes. We are confident that we are capturing the signal of module presence. We think this is a worthwhile topic for future exploration.

Additionally, in the first iteration of training data curation, we utilized an e-value cutoff of ≤1e-10 to retain all gene annotations at or below this threshold. We have since revised the training data, and now instead of using a consistent e-value cutoff of ≤1e-10 for our KofamScan-derived annotations, we updated our approach by utilizing the adaptive threshold HMM score cutoffs of the KofamScan HMM profiles. This approach provides resilience to using specific e-value cutoffs by preventing inflation of the training and assessment datasets with less-confident gene annotations. We now include a description of this revised process in our methods subsection “Filtering genome database metadata, downloading high-quality genomes, and gene annotations”.

2. The "KofamScan command line tool" was used to annotate the training set for this method. Of course, other annotation methods (e.g. DRAM) may render significant differences in the resulting functional annotations. Did the authors test if the KofamScan-trained classifiers show similar accuracy/performance when applied to annotations from DRAM or some other competing tool?

Without a KEGG license, DRAM internally runs KofamScan and reproduces the results that we generated using the KofamScan tool. As perhaps many of the potential users of this tool, we do not possess a KEGG license, and therefore could not compare DRAM’s alternative annotation method (that utilizes the KEGG GENES database) to KofamScan annotations.

To assess the quality of our KofamScan annotations, we re-annotated a subset of 25 training genomes using the BlastKoala (designed to annotate genomes) and GhostKoala (intended to annotate metagenomes) online annotation tools available from the KEGG database. The annotations from BlastKoala were ~85% identical, while KofamScan annotated ~4% of genes on average that were missed by BlastKoala, and BlastKoala labelled ~9% of genes that did not surpass any HMM adaptive thresholds from KofamScan. Approximately 2% of annotations differed between the two tools. The results were similar with GhostKoala: ~84% of annotations were identical, GhostKoala annotated 8% of genes that KofamScan missed; KofamScan labelled ~6% of genes that GhostKoala missed; ~2% of annotations were a mismatch. We are confident in using KofamScan for our annotations given the similarity of results yielded by GhostaKoala and BlastKoala on this subset of genomes.

3. Would it be possible to train the method on KEGG reaction IDs so other approaches that annotate reactions directly could be applied to the method? Not suggesting the authors do this work, but it might be worth mentioning in the discussion.

About 85% of reactions (n = 836) involved in the KEGG modules contained within MetaPathPredict are uniquely distributed in exactly one module. Roughly ~3.5% are contained within 3 or more modules (with a maximum number of 7 modules containing the same reaction). MetaPathPredict is essentially predicting the presence or absence of groups of KEGG reactions that have minimal overlap across KEGG modules, if at all. While there would be utility in training models to predict individual KEGG reactions, this is outside the scope of MetaPathPredict.

4. Did the authors look at the features chosen by the classifiers for various modules? It would be interesting to know how often the top-ranking features lie within the module or outside, and when outside, what kind of feature is used?

Thank you for your question. Please see our response to comment #1 from Reviewer #2.

5. The link in the PDF to the git repo (https://github.com/d-mcgrath/MetaPathPredict) is corrupted somehow and doesn't work. I suspect the "-". Manually entering the address does get me to the repo.

Thank you for bringing this to our attention. We have fixed the dash in the manuscript and tested the link – it appears to be functioning as expected now.

Reviewer #2 (Recommendations for the authors):Observation 1: Figures 2-5 present compelling evidence of MetaPathPredict's predictive capabilities. However, the authors have not discussed the potential mechanisms that could be discovered using the stacked ensemble of neural networks. In lines 171-173, the authors mentioned that MetaPathPredict's models incorporate information from genes outside of KEGG modules, but they have not elaborated on how such information can be interpreted. While predictive power is a commendable goal, machine learning models usually trade off explanatory power for predictive power. Therefore, the authors should discuss whether the stacked ensemble of neural networks could provide biological insights (https://doi.org/10.1038/s42256-019-0048-x).

Thank you for your comment. We have calculated the importance of both models’ features using the SHAP method (Lundberg and Lee, 2017), a mathematical method to explain the predictions of machine learning models. Features with large absolute SHAP values play an important role in calculating a model’s predictions. After deriving SHAP values for each feature from our models, we sorted the features by decreasing importance. SHAP scores in model #1 (trained to classify modules present in ≥10% and ≤90% of training genomes) indicate that 30 of the top 100 most important features (genes) influencing predictions were direct components of KEGG modules predicted by this model. In model #2 (classifies modules present in <10% or >90% of training genomes), 37 of the 100 most influential features were part of KEGG modules this model was trained to predict. Given the multilabel architecture of our models, it is difficult to draw direct conclusions from SHAP analysis. However, it is clear that MetaPathPredict’s predictions are at least in part influenced by select genes present in KEGG modules, and also to a larger extent by genes not directly participating in KEGG module reactions. We have now added a section that discusses this in our results subsection “Analysis of model feature importance using SHAP”, as well as our conclusion. We additionally provide a supplementary table 3 including all features, a description of their function, which modules they are directly involved in (if any), and their associated mean SHAP values. Interestingly, a number of these features were found to be transporters, sensing components, regulators, pathogenesis-related proteins, and transposases from mobile elements. We include a discussion of the potential biological implications of the important features in the Results and Conclusions section.

Observation 2: The authors have primarily focused on predicting the presence of complete KEGG modules, which may result in predictions that are overly conservative regarding gene essentiality. For example, some KEGG modules may have functionally redundant reactions, and an incomplete KEGG module could still lead to a viable set of metabolic reactions. To address this issue, the authors could predict gene essentiality, carbon source, or metabolic product. (For further information, see https://doi.org/10.1186/s13059-021-02295-1.)

Thank you for bringing this to our attention. To address the issue of reaction redundancy, we assessed the completeness KEGG modules using a list of all possible paths to completion for each one. This list was created by utilizing methods from the Anvi’o Python module (Eren et al. 2015), which calculates all possible non-redundant combinations of KEGG Orthologs (K numbers) required to complete a module. For a module to be considered “present” in the training data, only one set of K numbers needed for a non-redundant path to module completion had to be present in a genome, as opposed to all genes being required that could (in some cases redundantly) complete a module. We now include a revised description of the process in the methods subsection “Formatting gene annotation data, fitting KEGG module classification models.”

References

Lundberg, Scott M., and Su-In Lee. "A unified approach to interpreting model predictions." *Advances in neural information processing systems* 30 (2017).

Eren, A. Murat, et al. "Anvi’o: an advanced analysis and visualization platform for ‘omics data." *PeerJ* 3 (2015): e1319.